# Cannabis Use among Cancer Survivors amid the COVID-19 Pandemic: Results from the COVID-19 Cannabis Health Study

**DOI:** 10.3390/cancers13143495

**Published:** 2021-07-13

**Authors:** Marlene Camacho-Rivera, Jessica Y. Islam, Diane L. Rodriguez, Denise C. Vidot

**Affiliations:** 1Department of Community Health Sciences, School of Public Health, SUNY Downstate Health Sciences University, Brooklyn, NY 11203, USA; marlene.camacho-rivera@downstate.edu; 2Cancer Epidemiology Program, H. Lee Moffitt Cancer Center and Research Institute, Tampa, FL 33612, USA; jessica.islam@moffitt.org; 3Morsani College of Medicine, University of South Florida, Tampa, FL 33602, USA; dianer@usf.edu; 4School of Nursing and Health Studies, University of Miami, Coral Gables, FL 33146, USA; 5Sylvester Comprehensive Cancer Center, Miller School of Medicine, University of Miami, Miami, FL 33136, USA

**Keywords:** cannabis, marijuana, COVID-19, pandemic, cancer, substance use, chronic disease

## Abstract

**Simple Summary:**

During the COVID-19 pandemic, cancer survivors have been identified as a population with an increased risk for adverse outcomes. This represents an additional burden on cancer patients, who already need to cope with a variety of symptoms associated with their cancer diagnosis. In this study, we analyzed results from a COVID-19 cannabis health study and found that individuals with a history of cancer are more likely to report cannabis use to manage mental health and pain symptoms, and are more likely to report fear of a COVID-19 diagnosis, compared to adults without a history of cancer. These results support the importance and need for conversations between clinicians and their patients, particularly cancer survivors, about the use of cannabis.

**Abstract:**

Clinical indications for medicinal cannabis use include those with cancer, a subgroup advised to avoid exposure to COVID-19. This study aims to identify changes to cannabis use, methods of cannabis delivery, and coping strategies among cancer survivors since the pandemic by cancer status. Chi-squared tests were used for univariate comparisons of demographic characteristics, cannabis use patterns, COVID-19 symptoms, and coping behaviors by cancer survivor status. Data included 158 responses between 21 March 2020 and 23 March 2021, from medicinal cannabis users, categorized as cancer survivors (*n* = 79) along with age-matched medicinal cannabis users without a history of cancer (*n* = 79). Compared to adults without a history of cancer, cancer survivors were more likely to report use of cannabis as a way of managing nausea/vomiting (40.5% versus 20.3%, *p* = 0.006), headaches or migraines (35.4% versus 19.0%, *p* = 0.020), seizures (8.9% versus 1.3%, *p* = 0.029), and sleep problems (70.9% versus 54.4%, *p* = 0.033), or as an appetite stimulant (39.2% versus 17.7%, *p* = 0.003). Nearly 23% of cancer survivors reported an advanced cannabis supply of more than 3 months compared to 14.3% of adults without a history of cancer (*p* = 0.002); though the majority of cancer survivors reported less than a one-month supply. No statistically significant differences were observed by cancer survivor status by cannabis dose, delivery, or sharing of electronic vaping devices, joints, or blunts. Cancer survivors were more likely to report a fear of being diagnosed with COVID-19 compared to adults without a history of cancer (58.2% versus 40.5%, *p* = 0.026). Given the frequency of mental and physical health symptoms reported among cancer survivors, clinicians should consider conversations about cannabis use with their patients, in particular among cancer survivors.

## 1. Introduction

Since the declaration of COVID-19 as a global pandemic by the World Health Organization on 11 March 2020, disparities in COVID-19-associated morbidity and mortality have emerged among older adults and individuals living with chronic health conditions [1,2]. Within the United States, medicinal cannabis use is legal in certain states for individuals with certain health conditions, such as cancer [3]. Cancer survivors, including those who use medicinal cannabis, have been recommended to take additional precautions to reduce COVID-19 exposure due to increased risk for COVID-19 hospitalization and mortality among individuals with pre-existing chronic health conditions [4,5,6,7]. Beyond the physical risks of COVID-19, individuals with chronic health conditions are reporting increased anxiety and depressive symptoms, as well as adverse economic outcomes including loss of employment and health insurance [8,9,10]. Prior studies, including those conducted by the study team, have documented increased reports of physical and mental health symptoms during the pandemic period among cancer survivors compared to adults without a history of cancer [4,11].

Legal qualifying medical conditions for cannabis use vary by state within the United States; cancer and HIV/AIDs are conditions consistently included across all legal states [12]. Despite variation in the legality of cannabis, “conclusive or substantial evidence” in the effectiveness of cannabis to treat chronic pain, chemotherapy-induced nausea and vomiting, and patient-reported multiple sclerosis spasticity symptoms were identified in 2017 by the National Academies of Sciences, Engineering, and Medicine in a report on the Health Effects of Cannabis and Cannabinoids: The Current State of Evidence and Recommendations for Research [13]. Cannabis contains phytocannabinoids [i.e., cannabidiol (CBD) and tetrahydrocannabidiol (THC)] that have been shown to engage with the endocannabinoid system, an endogenous system responsible for homeostasis. Receptors for the endocannabinoid system, CB1 and CB2, are located throughout the body and bind with phytocannabinoids [14] depending on their affinity. CB2 receptors have been identified as opportunities for cannabis, CBD in particular, to impact the health of cancer patients due to its opportunity to effect pain and immune function, for example [14].

Though clinical evidence on the indications of cannabis use for cancer has traditionally been limited due to federal legalization status, observational studies of cannabis consumers identify pain management, nausea, appetite, anxiety, depression, and sleep as most prevalent reasons for use [3,15]. Further, medical cannabis use has been identified as a coping strategy to help manage physical (i.e., pain) and mental health symptoms (i.e., depression, stress) among cancer survivors [15,16,17,18,19]. However, the impacts of the COVID-19 pandemic on cannabis use behaviors and other coping strategies among cancer survivors remains underexplored. Cannabis as a coping strategy differs from other strategies largely due to its unique relationship with the human endocannabinoid system. Cannabis has the opportunity to impact physical and mental health through cannabinoids that engage with the body based on the type of cannabis used and the method of delivery. The objective of the study is to identify changes to cannabis use, methods of cannabis delivery, and coping strategies among cancer survivors, and to describe differences in cannabis behaviors and coping strategies by cancer status among an age-matched sample of cannabis users from the COVID-19 cannabis health study.

## 2. Materials and Methods

### 2.1. COVID-19 Cannabis Health Study

Cross-sectional data are from the COVID-19 cannabis health study, a multisite study designed to examine COVID-19 impacts on cannabis use patterns and related behaviors among adult medicinal and recreational cannabis users [11]. The study includes domains regarding COVID-19 testing, diagnosis, and probable COVID-19 symptoms; measures of mental health symptoms reported since the COVID-19 pandemic; cannabis and other substance use behaviors before and during the pandemic period; and self-report of physician diagnosis of chronic health conditions. Study eligibility criteria included adults 18 years of age or older; participants included cannabis users (both medicinal and/or recreational) and non-cannabis users.

The University of Miami Institutional Review Board approved the study. All participants provided informed consent electronically before beginning the survey. Data collection and management occurred within REDCap software hosted at the University of Miami. Data for this analysis included 158 responses from 21 March 2020 to 23 March 2021 from participants mostly residing in the United States (92%). Thirteen participants (8%) were from other countries including Canada, Colombia, France, Israel, Kenya, Malaysia, Mexico, and New Zealand. Participants were categorized as cancer survivors (*n* = 79) along with age-matched medicinal cannabis users without a history of cancer (*n* = 79).

### 2.2. Measures

The primary exposure for this analysis was cancer survivor status based on the participant’s response to two questions. Participants were categorized as a cancer survivor if they selected cancer as a response option of either of the following questions: “Which of the following conditions do you currently live with?” or “What conditions do you manage with your cannabis?”

The COVID-19 Cannabis Health Questionnaire, a 25-item measure administered to medicinal and recreational cannabis users, was used to obtain data regarding patterns of cannabis use, other substance use behaviors (alcohol, illicit drugs), and report of physical and mental health symptoms during the pandemic period [20]. Additional details regarding the COVID-19 Cannabis Health Questionnaire may be found in RTI International’s PhenX Toolkit (https://www.phenxtoolkit.org/covid19/, Accessed on 12 June 2021), an open access repository of recommended measurement protocols. Primary outcomes for this analysis were cannabis use behaviors since the start of the COVID-19 pandemic. It included questions around the dose, method of cannabis delivery, cannabinoid type within medical cannabis, and length of cannabis supply. COVID-19 questions included self-report of current COVID-19-related symptoms, COVID-19 testing, and fear of COVID-19 diagnosis or transmission to others. Behaviors used to cope with the COVID-19 pandemic included mediation, eating or physical activity, sleep habits, talking to friends and family or health care professionals, cannabis use, and sexual activity. Substance use questions included use (yes/no) and change in frequency of the use of tobacco, alcohol, opioids, methamphetamines, cocaine, and psilocybin.

### 2.3. Data Analysis

Chi-squared tests among the age-matched sample were used for univariate comparisons of categorical variables including demographic characteristics, cannabis use patterns, COVID-19 symptoms, and coping behaviors by cancer survivor status. All statistical analyses were conducted in SAS v9.4 (SAS Institute, Cary, NC, USA). The Type I error was maintained at 5%.

## 3. Results

### 3.1. Demographics

The majority of the participants (145 or 92%) were residents of the United States, while thirteen participants (8%) were residents of other countries including Canada, Colombia, France, Israel, Kenya, Malaysia, Mexico, and New Zealand. Of the cancer patients surveyed, 51% were female, 47% were male, and 1% identified as Other, with the mean age of respondents being 58.1 years. Eighty-two percent identified as white, 9% as Hispanic, 5% as black or African American, 1% as Asian, and 3% as other. Comorbidities were reported by respondents, including 44%, 30%, and 13% of cancer patients reporting mental health, cardiometabolic, and respiratory comorbidities, respectively. Ninety-six percent of cancer patients identified as medical cannabis users (Table 1).

### 3.2. Health Conditions and Symptoms Managed by Cannabis Users

Figure 1 illustrates the frequency of health conditions managed by cannabis users across the sample. Among respondents without a history of cancer, 67% of respondents used cannabis to manage anxiety symptoms, 49% for chronic pain, 42% used cannabis to manage depressive symptoms, 28% for post-traumatic stress disorder (PTSD) symptoms, 8% for an autoimmune disease, and 11% for irritable bowel syndrome symptoms. Among cancer survivors, 61% of respondents used cannabis to manage anxiety symptoms, 54% for chronic pain, 48% used cannabis to manage depressive symptoms, 25% for PTSD symptoms, 11% for another autoimmune disease, and 15% for irritable bowel syndrome symptoms. These results were not considered statistically significant from one another.

Table 2 describes the prevalence of symptoms reported that were managed by cannabis among cancer survivors and those without a history of cancer. Compared to participants without a history of cancer, cancer survivors were more likely to report use of cannabis to manage nausea/vomiting (40.5% versus 20.3%, *p* = 0.006), as an appetite stimulant (39.2% versus 17.7%, *p* = 0.003), to manage headaches or migraines (35.4% versus 19.0%, *p* = 0.020), seizures (8.9% versus 1.3%, *p* = 0.029), and sleep problems (70.9% versus 54.4%, *p* = 0.033). Adults without a cancer history were more likely to report use of cannabis without management of any symptoms compared to cancer survivors (8.9% versus 0%, *p* = 0.007).

### 3.3. Cannabis Use Behaviors and COVID-19-Related Concerns

Table 3 describes self-reported cannabis use behaviors since the onset of the pandemic by cancer survivor status. Compared to those without a history of cancer, cancer survivors were significantly more likely to report a longer advanced supply of cannabis. Nearly 23% of cancer survivors reported an advanced supply of more than 3 months compared to 14.3% of adults without a history of cancer (*p* = 0.002). Cancer survivors were more likely to report that a health professional recommended that they obtain an advanced supply of cannabis (15.9%) compared to adults without cancer (4.0%) (*p* = 0.039). Thirty-two percent of cancer survivors reported that the dose of cannabis use had increased since COVID-19 pandemic onset. No statistically significant differences were observed by cancer survivor status by cannabis dose, methods of cannabis delivery, or sharing of electronic vaping devices or joints, blunts, or spliffs. No statistically significant differences in cannabis use, other coping strategies, or COVID-19 concerns emerged across demographic characteristics (e.g., place of residence, sex). Differences in self-report of other substances (tobacco, alcohol, opioids, methamphetamines, cocaine, and psilocybin) and other coping strategies were not statistically significant by cancer survivor status. Details are presented in Appendix A.

As shown in Figure 2, self-report of COVID-19 testing, symptoms, receipt of COVID-19 diagnosis, fear of COVID-19 transmission or reports of isolation due to COVID-19 were not statistically significant by cancer survivor status. Cancer survivors were significantly more likely to report fear of a COVID-19 diagnosis compared to adults without a history of cancer (58.2% versus 40.5%, *p* = 0.026).

## 4. Discussion

This study summarizes differences by cancer history in cannabis use patterns, changes in the method of delivery, reports of physical and mental health symptoms managed by cannabis, and COVID-19-related fears among an age-matched sample of respondents from the COVID-19 cannabis health study. Findings from the study suggest that cancer survivors are frequently reporting the use of cannabis to manage both physical and mental health symptoms associated with their cancer diagnosis, which is consistent with studies of cannabis use among cancer survivors before the COVID-19 pandemic [21,22,23,24]. Specifically, self-reported symptoms most frequently managed by medicinal cannabis among respondents included anxiety and pain. While no differences in frequency of cannabis use or method of delivery were observed between those with and without a cancer diagnosis, cancer survivors were more likely to have an advanced supply of cannabis. However, most cancer survivors (77%) reported an advanced supply of less than one month.

Similar to our findings, other studies have also identified anxiety and sleep problems as primary symptoms managed by cancer patients using medicinal cannabis [25,26,27,28]. 

Opportunities exist for cannabis use as a form of palliative care and non-curative treatment among cancer patients. Early intervention with palliative care among cancer patients can increase overall survival and improve quality of life of both cancer patients and their caregivers [29]. Integrating medicinal cannabis into palliative care can address disparities and underuse of palliative care among cancer patients due to poor effectiveness of current palliative treatment options, as indicated by poor symptom control and intolerable adverse effects attributed to palliative treatment options such as opioids [30]. Anxiety and depression are common mental health symptoms experienced by cancer patients. These mental health symptoms can manifest as worry and poor sleep quality, and about one-third of cancer patients experience psychological distress that requires clinical treatment, although this proportion varies greatly by cancer type and prognosis [31]. As the endocannabinoid system is involved in mood regulation, cannabis-derived treatment could improve mental health symptoms, as has been shown in mice models [32,33]. Curative cancer treatment can also cause physical symptoms such as nausea, vomiting, pain, and neuropathy. Modern antiemetic regimens are less effective at controlling nausea with 40–70% of patients reporting nausea while receiving highly or moderately emetogenic chemotherapy. Chemotherapeutic agents induce nausea and vomiting through elevated release of serotonin which bind to the 5-hydroxytryptmaine 3 (5-HT3)receptors, which send information of excess chemicals to the brain and directly promote emesis [18]. Cannabinoids can directly inhibit these receptors and are thought to act as modulators and indirect agonists on the autoreceptors of the 5-HT3 [34]. The bioactive benefits of cannabis may outweigh its risks, particularly in the context of palliative care. Future research should focus on providing further evidence on the mechanistic pathways and effectiveness to optimize medicinal cannabis applications and dosage.

The COVID-19 pandemic has led to added stress and anxiety among cancer patients [35]. We observed that cancer survivors were more likely to fear a COVID-19 diagnosis compared to those without a history of cancer. Higher levels of fear of contracting COVID-19 among cancer survivors are likely due to both early and more recent reports of increased risk of COVID-19 morbidity and mortality among adults with pre-existing conditions, including cancer [36,37,38,39,40,41,42,43]. To cope with the psychological impacts of the pandemic, cancer patients are exhibiting several coping strategies. For example, US-based ovarian cancer patients reported using emotional support, self-care, hobbies, planning, positing reframing, and religion as coping strategies [44]. Older cancer patients in the US reported to engage in physical activities such as gardening, walking, fitness regimes, tai chi, yoga, and fishing to stay busy or active during the pandemic [45]. Similarly, respondents to our survey reported physical activity as a common coping mechanism. We observed no differences in reported coping strategies by cancer survivor status, which may be explained by the increased use of cannabis for non-medical purposes across the U.S. [46,47].

The results of this study should be considered in light of the limitations of the study. First, due to the cross-sectional nature of the study, causation and temporality cannot be determined. Due to the anonymity of survey responses, there may be repeat responses, although data cleaning and reCAPTCHA methods in REDCap were used to avoid multiple responses. In addition, no monetary or other incentives were provided, thus reducing the likelihood of intentional repeated responses. Generalizability of study findings may be limited due to the electronic nature of the survey, which excludes cannabis users without internet access. Furthermore, due to the self-reported nature of the data, there is potential for recall bias and misclassification bias of COVID-19 and cannabis behaviors and symptoms. Additionally, due to the urgent need for data collection early in the pandemic, the COVID-19 Cannabis Questionnaire was not validated within the target population prior to dissemination in the field. Medicinal cannabis use was based on self-report without medical record or prescription confirmation. Lastly, while differences in the age distribution between cancer survivors and non-cancer survivors were considered in our matching strategy, we describe bivariate associations between cannabis behaviors, COVID-19 symptoms, coping strategies by cancer status, and additional factors including race/ethnicity, gender, and socioeconomic status, which may explain the differences observed in the sample.

## 5. Conclusions

Overall, we observed that cancer survivors are frequently reporting the use of cannabis to manage both physical and mental health symptoms associated with their cancer diagnosis and that cancer survivors are more likely to report fear of a COVID-19 diagnosis compared to those without a history of cancer. Currently, medicinal cannabis is not clinically indicated for the management of anxiety or depression in most states with legalized medicinal cannabis. Additional studies are needed to examine potential positive and/or negative implications of cannabis use on SARS-CoV-2 risk, transmission, morbidity, and mortality. Given the frequency of mental and physical health symptoms reported among cancer survivors during the COVID-19 pandemic period, clinician–patient interactions should include questions around cannabis use, particularly those with a history of cancer. Practically, education campaigns on the endocannabinoid system, phytocannbinoids (i.e., CBD, THC), and reasons why cannabis impacts health among cancer survivors should be designed and tested to improve the medicinal use of cannabis among cancer patients.

## Figures and Tables

**Figure 1 cancers-13-03495-f001:**
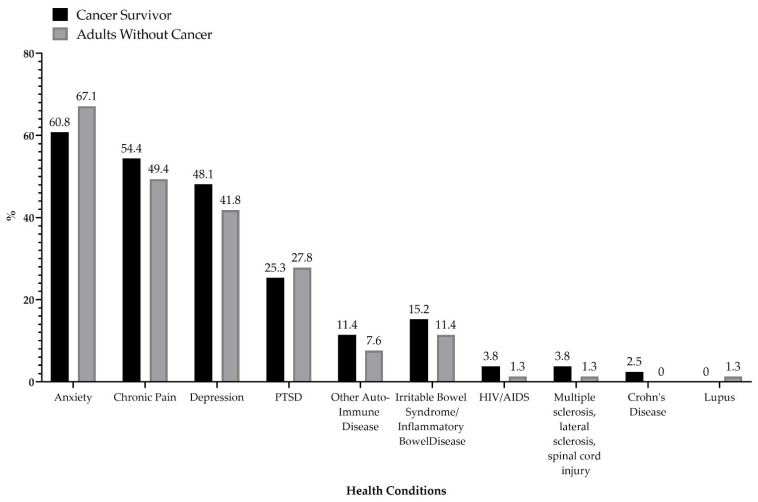
Frequency of Health Conditions Managed with Cannabis among the COVID-19 Cannabis Health Questionnaire respondents (21 March 2020–23 March 2021) among age-matched cancer survivors (*n* = 158).

**Figure 2 cancers-13-03495-f002:**
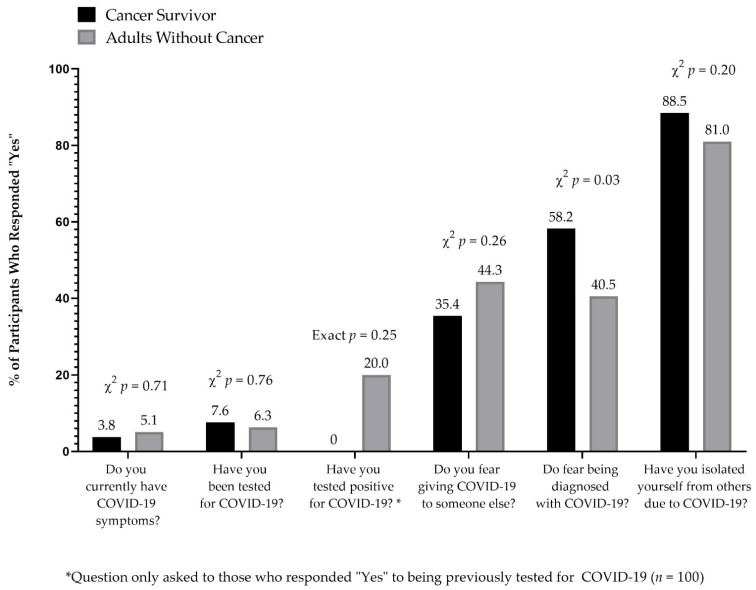
COVID-19-Related Concerns among the COVID-19 Cannabis Health Questionnaire respondents (21 March 2020–23 March 2021) among age-matched cancer survivors (*n* = 158).

**Table 1 cancers-13-03495-t001:** Demographic Characteristics of COVID-19 Cannabis Health Questionnaire Respondents (21 March 2020 to 23 March 2021) among age-matched Cancer Survivors (*n* = 158).

	Total		Adults Without Cancer		Cancer Survivors		*p*-Value
	*n*	%	*n*	%	*n*	%	
What is your gender?							0.109
Female	68	43.3	28	35.4	40	51.3	
Male	87	55.4	50	63.3	37	47.4	
Other	1	0.6	0	0	1	1.3	
Transgender	1	0.6	1	1.3	0	0	
Do you live in the United States?							0.043
No	13	8.2	3	3.8	10	12.7	
Yes	145	91.8	76	96.2	69	87.3	
Race/Ethnicity							0.427
Non-Hispanic White	132	85.2	69	88.5	63	81.8	
Non-Hispanic Black or African American	6	3.9	2	2.6	4	5.2	
Hispanic	9	5.8	2	2.6	7	9.1	
Non-Hispanic Asian	2	1.3	1	1.3	1	1.3	
Native Hawaiian or Other Pacific Islander	1	0.6	1	1.3	0	0	
American Indian or Alaska Native	5	3.2	3	3.8	2	2.6	
Education Level							0.983
High school or less	18	11.4	9	11.4	9	11.4	
Bachelor’s degree or some college	99	62.7	50	63.3	49	62	
Masters degree or higher	41	25.9	20	25.3	21	26.6	
Household Income							0.876
Less than $30,000	38	32.2	19	31.1	19	33.3	
Between $30,000 and $50,000	29	24.6	16	26.2	13	22.8	
Between $50,000 and $100,000	26	22	12	19.7	14	24.6	
More than $100,000	25	21.2	14	23	11	19.3	
Medical Cannabis User							0.000
No	24	15.2	21	26.6	3	3.8	
Yes	134	84.8	58	73.4	76	96.2	
Employed for Wages							0.714
No	118	74.7	58	73.4	60	75.9	
Yes	40	25.3	21	26.6	19	24.1	
Self-Employed							0.428
No	126	79.7	65	82.3	61	77.2	
Yes	32	20.3	14	17.7	18	22.8	
Unemployed Due to COVID							0.118
No	147	93	71	89.9	76	96.2	
Yes	11	7	8	10.1	3	3.8	
Unemployed Prior to COVID							0.119
No	141	89.2	73	92.4	68	86.1	
Yes	17	10.8	6	7.6	11	13.9	
Respiratory Comorbidities							0.198
No	132	83.5	63	79.7	69	87.3	
Yes	26	16.5	16	20.3	10	12.7	
Cardiometabolic Comorbidities							0.478
No	114	72.2	59	74.7	55	69.6	
Yes	44	27.8	20	25.3	24	30.4	
Mental Health Comorbidities							0.749
No	86	54.4	42	53.2	44	55.7	
Yes	72	45.6	37	46.8	35	44.3	
Pain							0.749
No	88	55.7	45	57	43	54.4	
Yes	70	44.3	34	43	36	45.6	

**Table 2 cancers-13-03495-t002:** Self-Reported Symptoms Managed Using Cannabis of the COVID-19 Cannabis Health Questionnaire respondents (21 March 2020–23 March 2021) among Age-Matched Cancer Survivors (*n* = 158).

	Total		Adults Without Cancer		Cancer Survivors		*p*-Value
Symptoms	*n*	%	*n*	%	*n*	%	
Acute Pain	59	37.3	29	36.7	30	38	0.869
Chronic non-cancer pain	74	46.8	34	43	40	50.6	0.339
Cancer pain	33	20.9	0	0	33	41.8	< 0.001
Nausea/vomiting	48	30.4	16	20.3	32	40.5	0.006
Appetite Stimulant	45	28.5	14	17.7	31	39.2	0.003
Headaches/migraines	43	27.2	15	19	28	35.4	0.02
Muscle spams	44	27.8	22	27.8	22	27.8	1
Seizures	8	5.1	1	1.3	7	8.9	0.029
Sleep problems	99	62.7	43	54.4	56	70.9	0.033
Alcohol withdrawal symptoms	7	4.4	3	3.8	4	5.1	0.69
Opioid withdrawal symptoms	8	5.1	2	2.5	6	7.6	0.147
Palliative care	8	5.1	2	2.5	6	7.6	0.147
Other	20	12.7	9	11.4	11	13.9	0.632

**Table 3 cancers-13-03495-t003:** Cannabis Use Patterns During the COVID-19 Pandemic Among the COVID-19 Cannabis Health Questionnaire respondents (21 March 2020–23 March 2021) among age-matched cancer survivors (*n* = 158).

	Total		Adults Without Cancer		Cancer Survivors		*p*-Value
	*n*	%	*n*	%	*n*	%	
Have you had any pain in your throat, chest, or lungs after using cannabis?							0.344
No	133	84.2	65	82.3	68	86.1	
Yes	25	15.8	14	17.7	11	13.9	
Since COVID-19 has been declared a pandemic, are you (or were you) worried about not being able to pay for your cannabis?							0.012
No	88	58.7	51	68.9	37	48.7	
Yes	62	41.3	23	31.1	39	51.3	
Since COVID-19 has been declared a pandemic, did a health professional recommend you use cannabis to manage COVID-19 or the coronavirus?							0.311
No	154	97.5	78	98.7	76	96.2	
Yes	4	2.5	1	1.3	3	3.8	
Since COVID-19 has been declared a pandemic, did a health professional recommend you to get an advance supply of your cannabis?							0.039
No	106	89.1	48	96	58	84.1	
Yes	13	10.9	2	4	11	15.9	
Since COVID-19 has been declared a pandemic, how has the dose of your cannabis use changed?							0.483
The amount used has increased	43	28.7	19	25.7	24	31.6	
The amount used has decreased	18	12	11	14.9	7	9.2	
The amount used has stayed the same	89	59.3	44	59.5	45	59.2	
Since COVID-19 has been declared a pandemic, do you share joints, blunts, or spliffs?							0.999
Yes, more than usual	2	1.3	1	1.4	1	1.3	
Yes, the same as usual	16	10.7	8	10.8	8	10.5	
Yes, less than usual	4	2.7	2	2.7	2	2.6	
No, I did not share before COVID-19	88	58.7	44	59.5	44	57.9	
No, I stopped	40	26.7	19	25.7	21	27.6	
Since COVID-19 has been declared a pandemic, do you share electronic vaporizing devices?							0.573
Yes devices containing cannabis	11	7	4	5.1	7	8.9	
Yes devices containing nicotine							
Yes devices containing both cannabis and nicotine							
No I do/did not share devices	91	57.6	45	57	46	58.2	
No I do/did not use devices	56	35.4	30	38	26	32.9	
What is the dominant cannabinoid (i.e., THC, CBD, CBN) within your medical cannabis?							0.257
CBD dominant	7	5	2	2.9	5	6.8	
CBN dominant	3	2.1	0	0	3	4.1	
THC dominant	87	61.7	45	66.2	42	57.5	
CBD and THC ratio	38	27	17	25	21	28.8	
Other cannabinoid dominant							
Unsure	6	4.3	4	5.9	2	2.7	
How often were you under the influence of psychoactive cannabis for 6 or more hours?							0.587
Never	16	10.3	8	10.3	8	10.4	
Less than Monthly	11	7.1	4	5.1	7	9.1	
Monthly	4	2.6	3	3.8	1	1.3	
Weekly	27	17.4	16	20.5	11	14.3	
Daily/Almost Daily	97	62.6	47	60.3	50	64.9	
Does your health insurance cover the cost of your medical cannabis?							0.563
Yes							
No	111	92.5	48	94.1	63	91.3	
I do not have health insurance	9	7.5	3	5.9	6	8.7	
How long will the advance supply of cannabis last you?							0.001
1 week	3	4.2	3	8.3	0	0	
2 weeks	18	25.4	5	13.9	13	37.1	
3 weeks	8	11.3	4	11.1	4	11.4	
1 month	15	21.1	5	13.9	10	28.6	
2 months	10	14.1	10	27.8	0	0	
3 months	3	4.2	3	8.3	0	0	
More than 3 months	14	19.7	6	16.7	8	22.9	
Before COVID-19 has been declared a pandemic, which method of cannabis delivery did you use most?							0.18
Smoked it in a pipe/bowl	6	30	2	25	4	33.3	
Smoked it in a blunt	3	15	1	12.5	2	16.7	
Smoked it in a joint	3	15	0	0	3	25	
Edible (in food or drink)	3	15	3	37.5	0	0	
Vaporizer	3	15	2	25	1	8.3	
Ointment, cream, patch							
Tincture	1	5	0	0	1	8.3	
Pill	1	5	0	0	1	8.3	
Since COVID-19 has been declared a pandemic, which method of delivery do you use?							0.67
Smoked it in a pipe/bowl	5	25	2	25	3	25	
Smoked it in a blunt							
Smoked it in a joint	3	15	1	12.5	2	16.7	
Edible (in food or drink)	4	20	1	12.5	3	25	
Vaporizer	4	20	3	37.5	1	8.3	
Ointment, cream, patch							
Tincture	3	15	1	12.5	2	16.7	
Pill	1	5	0	0	1	8.3	

CBD—Cannabidiol; CBN—Cannabinol; THC—Tetrahydrocannabinol

## Data Availability

The data presented in this study are available on request from the corresponding author.

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
