# Peer review of "Cannabis Use among Cancer Survivors amid the COVID-19 Pandemic: Results from the COVID-19 Cannabis Health Study"

_cancers, 2021, doi:10.3390/cancers13143495_

Round 1

Reviewer 1 Report

This study investigated changes to cannabis use, methods of cannabis delivery, and coping strategies among cancer survivors since the pandemic by cancer status. This is a very good work, with good methodology and well written manuscript.

Three Comments:

  1. Please give full name of PTSD (line 127).
  2. There are 13 (8%) participants who were not from USA. Are there any significant different in Cannabis use, coping strategies, etc. between these participants and those from USA?
  3. 43.3% of the participants are female and 55.4% are male. Are there any significant difference in Canabis use, coping strategies, fear of a COVID-19 diagnosis, etc. between female and male?

Author Response

Thank you kindly for the review of our manuscript. 

Reviewer 1 Comments

  1. Please give full name of PTSD (line 127).

Response: The full name has been included next to use of the acronym PTSD where it first appears in the manuscript.

  1. There are 13 (8%) participants who were not from USA. Are there any significant different in Cannabis use, coping strategies, etc. between these participants and those from USA?

Response: No statistically significant differences in cannabis use, coping strategies, or COVID-19 outcomes emerged by place of residence. This was included in results section 3.3.

  1. 43.3% of the participants are female and 55.4% are male. Are there any significant difference in Cannabis use, coping strategies, fear of a COVID-19 diagnosis, etc. between female and male?

Response: No statistically significant differences emerged in cannabis use, coping strategies, or COVID-19 outcomes emerged between men and women. This was included in results section 3.3.

Reviewer 2 Report

The manuscript was prepared very well. The introduction section justifies the purpose of the study. I congratulate the authors for the preparation of the manuscript.

However, I have the following comments:

Introduction

  • In what other health conditions is cannabis authorized in the USA? Is a medical prescription necessary?
  • Please indicate what properties does cannabis have for cancer patients and is it indicated in all types of cancer, in which stages of the patients?
  • “Medical cannabis use has been identified as a coping strategy to help manage physiological and mental symptoms among cancer survivors”, which ones, how? What differentiates the cannabis strategy from other strategies for managing physiological and mental symptoms among cancer survivors?

Methodology

is generally correct. Are the questionnaires used validated?

Results

Tables and figures should have titles and figure/table captions that indicate more information about the content of the tables and figures.

Discussion

Line 179. It indicates several studies but only provides a reference. it should also discuss more precise arrears and not only indicate that anxiety. Clarify this.

Lines180-188. You provide a description of the results but do not compare with other coping strategies in cancer patients during the COVID-19 pandemic. could you comment?

Could the results of the study have any practical application to improve the use of cannabis in this type of patients?

Author Response

Thank you kindly for the review of our manuscript. 

Reviewer 2 Comments

Introduction

  • In what other health conditions is cannabis authorized in the USA? Is a medical prescription necessary?
  • Please indicate what properties does cannabis have for cancer patients and is it indicated in all types of cancer, in which stages of the patients?
  • “Medical cannabis use has been identified as a coping strategy to help manage physiological and mental symptoms among cancer survivors”, which ones, how? What differentiates the cannabis strategy from other strategies for managing physiological and mental symptoms among cancer survivors?

Response: We thank the reviewer for these questions and feedback. We have addressed these issues by greatly expanding upon the introduction to provide more background on medicine cannabis use.

Methodology

is generally correct. Are the questionnaires used validated?

Response: We appreciate the reviewer’s question. Like most COVID-19 questionnaires, due to the urgency of the pandemic and need for rapid data collection, the COVID-19 Cannabis Questionnaire was not validated prior to use. However, it has been included in the open access repository of the PhenX toolkit as one of the recommended measures for COVID-19. Nonetheless, we have included lack of questionnaire validation as a study limitation in the discussion section.

Results

Tables and figures should have titles and figure/table captions that indicate more information about the content of the tables and figures.

Response: We have updated the titles and captions of tables and figures to be more informative about the nature of their contents.

Discussion

Line 179. It indicates several studies but only provides a reference. it should also discuss more precise arrears and not only indicate that anxiety. Clarify this.

Response: We have updated the discussion to clarify this statement and provide additional references.

Lines180-188. You provide a description of the results but do not compare with other coping strategies in cancer patients during the COVID-19 pandemic. could you comment?

Response: We have significantly expanded the discussion section to provide a more in-depth description (along with citations) of published studies examining coping strategies among cancer patients during the COVID-19 pandemic.

Could the results of the study have any practical application to improve the use of cannabis in this type of patients?

Response: We have expanded the discussion section to provide additional recommendations to improve the use of cannabis among cancer patients.

Round 2

Reviewer 2 Report

The authors have completed all suggestions. 

I have nothing more to add.